# Hyperspectral Imaging as Powerful Technique for Investigating the Stability of Painting Samples

**DOI:** 10.3390/jimaging5010008

**Published:** 2019-01-03

**Authors:** Giuseppe Bonifazi, Giuseppe Capobianco, Claudia Pelosi, Silvia Serranti

**Affiliations:** 1Department of Chemical Engineering Materials & Environment, Sapienza, Rome University, Via Eudossiana 18, 00184 Rome, Italy; 2Department of Economics, Engineering, Society and Business Organization, Laboratory of Diagnostics and Materials Science, University of Tuscia, Largo dell’Università, 01100 Viterbo, Italy

**Keywords:** Hyperspectral imaging, painting samples, retouching pigments, watercolours, multivariate analysis

## Abstract

The aim of this work is to present the utilization of Hyperspectral Imaging for studying the stability of painting samples to simulated solar radiation, in order to evaluate their use in the restoration field. In particular, ready-to-use commercial watercolours and powder pigments were tested, with these last ones being prepared for the experimental by gum Arabic in order to propose a possible substitute for traditional reintegration materials. Samples were investigated through Hyperspectral Imaging in the short wave infrared range before and after artificial ageing procedure performed in Solar Box chamber under controlled conditions. Data were treated and elaborated in order to evaluate the sensitivity of the Hyperspectral Imaging technique to identify the variations on paint layers, induced by photo-degradation, before they could be detected by eye. Furthermore, a supervised classification method for monitoring the painted surface changes, adopting a multivariate approach was successfully applied.

## 1. Introduction

Hyperspectral imaging (HSI) is a diagnostic tool deserving great interest in the field of cultural heritage due to its non-invasive character and to the possibility of obtaining a lot of information with a single technique [1,2,3]. If coupled with chemometric techniques, it allows for gathering qualitative and/or quantitative information on the nature and physical-chemical characteristics of the investigated materials, and to combine imaging with spectroscopy for evaluating the distribution of materials on the surfaces [4,5,6,7,8,9]. By using classification methods, already applied in other research fields, it is possible to create a predictive model that is able to identify little variations of the painting layers due to the degradation phenomena of the constituent materials [10,11,12,13,14,15]. In conservation of cultural heritage, these classification methods could have great relevance because they allow to monitor in real time the surface changes by observing the spectra variation in respect to the calibration dataset. For these reasons, in the present work, HSI was applied with the aims to evaluate the sensitivity of the technique in order to identify the variations on paint layers, induced by photo-degradation, before they could be observed by eye and to use, following a multivariate based approach, the supervised classification methods for monitoring the painted surface changes [8,9]. As paint samples, a set of commercial watercolours was chosen together with various powder pigments, mainly iron oxide based materials, which were mixed with gum Arabic, without any additive, in order to verify their possible use in painting retouching. Iron oxide based pigments were chosen, as they are stable and widely used for millennia thanks to their durability [16,17,18]. However, when combined with gum Arabic and additives in commercial watercolours, the stability of paintings seems to be not the same [19,20]. The choice of testing watercolours derived from their wide use as materials for painting retouching, together with other products more or less recently introduced in the conservation field [21,22]. Watercolours are frequently used for retouching, especially by Italian conservators that, in particular, commonly choose Winsor&Newton as the preferred brand [20]. Watercolours are produced by the combination of a pigment with gum Arabic and other substances not specified by the manufacturer to safeguard the industrial patent [23,24,25,26,27,28]. The necessity to investigate the stability of retouching products is linked to the unknown and unpredictable behaviour of the commercial mixtures whose composition is not declared by suppliers [19,29,30,31,32]. Though watercolours are widely used in conservation, their stability in the long run has not been sufficiently studied [33,34,35,36,37,38] or it is limited to the investigation of pigment modification without examining the binder behaviour [39]. In general, even if retouching is a consolidated praxis in restoration, the monitoring of behaviour of retouched artworks is not widely applied, especially due to the high costs or lack of maintenance programs. However, several cases of chromatic alteration in areas retouched through watercolours were found, especially in red and brown painting zones where iron based pigments were used [20].

The evaluation of stability of commercial products, used in conservation, can be performed through different analytic and diagnostic techniques, requiring the preparation of a lot of micro-samples to perform the analyses [40]. The use of sampling based techniques is not always possible in conservation and monitoring, especially due to the difficulty or impossibility to repeat the measurements in the same points during the time. For this reason, non-invasive no-contact methods were chosen to study and monitor the photo degradation processes in watercolours and pigment powders. Specifically, HSI in the short ware infrared region (SWIR) was used with the aim to early detect and monitor the degradation of the investigated painting materials. HSI techniques were widely applied for the identification and characterization of paint layers but rarely for monitoring of degradation patterns [41,42,43,44,45,46,47]. Infrared reflectance spectroscopy is a well-known technique to obtain materials characterization and to set up a correct diagnostic plan based on non-destructive and non-invasive approach [4,48,49,50,51,52,53,54,55]. In particular, the SWIR range provides information about vibrational transitions, which are mostly overtones and combination bands whose fundamental transitions occur in the mid-IR. These features are often related to functional groups, like hydroxyl (−OH), carbonate (–CO_3_), and sulphate (–SO_4_) [56,57]. Absorption features from organic materials, like the paint binders, can also be observed and used to map their spatial distribution [44,58,59].

Based on a previously published paper, the present work extends the HSI results to the entire set of painting samples (totally 58) in order to make a comparison on a larger number of pigments [60].

In Section 2 (Materials and Methods), the experimental procedure will be reported. It is organized as follows: Section 2.1 sample preparation and ageing; Section 2.2 hyperspectral imaging (HSI), describing equipment and acquisition modalities; and, Section 2.3 spectral analysis, which reports in detail the data elaboration and the definition of prediction model used in the work. The Section 3 concerns results and shows spectra, PCA score plots, and prediction models. Section 4 reports discussion of the results shown in the Section 3. Last, Section 5 is devoted to the conclusions of the paper and to further possible research lines to develop.

## 2. Materials and Methods

### 2.1. Sample Preparation and Ageing

As painting materials, commercial watercolours, professional series supplied by Winsor&Newton both in form of tubes and pans, were selected in order to compare their stability, to light and UV radiation, with that of iron oxide pigments [61] supplied by Chroma with the specification of the country of origin, and two blue pigments in powder, all applied by gum Arabic (GA, by W&N) as binder in order to have the same binder of commercial watercolours (see Table 1 and Table 2 for pigment abbreviation and description). Only cobalt blue and ultramarine blue pigments were supplied by Zecchi (Florence).

In the case of natural iron oxide pigments, more than one colour typology was found, for this reason multiple samples are available for each kind of materials, i.e., five powders for burnt umber, nine powders for dark yellow ochre, etc. (Table 2). Commercial watercolours were chosen in order to have, for each colour and when available, two typologies: tube and pan, which were chosen due to their wide use in retouching (Table 1) [20]. The choice of samples for stability testing was also made on the base of previous data reported in experimental theses [18,20].

According to these data, the traditional watercolours that were used by conservators were classified between the less stable mixtures. For this reason, it was chosen to test the possibility of substituting these watercolours with materials having the same or similar colour appearance but prepared with natural pigments in powder and gum Arabic, without any additive.

The above described painting materials were homogeneously applied by brush on traditional gypsum/glue ground in order to create colour check tables with the different chosen pigments. Gypsum ground was chosen in order to simulate a painting repair to be covered by retouching, as commonly occurs in the practice of restoration of painting lacuna (Figure 1) [20].

Artificial ageing was performed by a model 1500E Solar Box chamber (Erichsen Instruments GmbH&Co, Hemer, Germany) simulating sunlight irradiation (visible and ultraviolet). The system is equipped with a 2.5 kW xenon-arc lamp and UV filter that cuts off the spectrum at 280 nm [62,63]. The samples were exposed in the Solar Box chamber from 1 to 504 h at 550 W/m^2^, 55 °C, and the UV filter at 280 nm. In these conditions, ageing was performed, evaluating the effects of light and UV radiation without considering other environmental agents, such as relative humidity. Inside the Solar Box chamber, relative humidity was constant (50%) and determined by the irradiation conditions. Relative humidity was monitored by a data logger positioned inside the Solar Box.

Hyperspectral imaging data were acquired at the following ageing times: 0 h, 168 h, 336 h, and 504 h, corresponding to 0 J/m^2^, 3.3 × 10^8^ J/m^2^, 6.7 × 10^8^ J/m^2^, and 1.0 × 10^9^ J/m^2^, respectively, of the total energy on the irradiated surfaces at the different times.

### 2.2. Hyperspectral Imaging (HSI)

Hyperspectral analyses were carried out on sample table at 0, 168, 336, and 504 h of exposure in the wavelength interval 1000–2500 nm (SWIR). The acquisitions were performed utilizing the SISUChema XL^TM^ (Specim, Oulu, Finland) device, equipped with a 31 mm lens allowing the acquisition of the paint layer with a resolution of 300 micron/pixel.

The spectral resolution was 6.3 nm. Illumination was obtained by SPECIM’s diffuse line illumination unit. Images were acquired through scanning each investigated sample line by line. Instrument is delivered with spectral calibration. Image data is automatically calibrated to reflectance by measuring an internal standard reference target before each sample scan.

The image correction was thus performed, adopting the following equation:(1)$$I=\frac{I_{0}-B}{W-B}\times 100,$$
where *I* is the corrected hyperspectral image, *I*_0_ is the original hyperspectral image, *B* is the black reference image (~0% reflectance), and *W* is the white reference image (~99.9% reflectance).

### 2.3. Spectral Analysis

HSI derived spectral data were analyzed by adopting standard chemometric methods [64,65], with the PLS_Toolbox (Version 8.2 Eigenvector Research, Inc., Manson, WA, USA) running inside Matlab (Version 8.4, The Mathworks, Inc., Natick, WA, USA). More in details, the spectra preprocessing was performed as follows: raw spectra were preliminary cut, at the beginning and at the end of the investigated wavelength range, in order to eliminate unwanted effects due to lighting/background noise. Preprocessing was adopted in order to reduce the noise and emphasize the spectral signal [66,67,68,69]. The following preprocessing algorithms were applied: standard normal variate (SNV) to reduce the effect of light scattering; 1st derivative to emphasize the spectral absorption of the investigated paint layers. Finally, Mean Center (MC) was adopted for centering the data before applying principal component analysis.

Principal Component Analysis (PCA) was applied as a powerful and versatile method that is capable of providing an overview of complex multivariate data. PCA can be used for revealing relations existing between variables and samples (e.g., clustering), detecting outliers, finding and quantifying patterns, generating new hypotheses, as well as many other things [70]. In this work, PCA was used to decompose the “processed” spectral data into several principal components (PCs), embedding the spectral variations of each collected spectral data set. The first few PCs, resulting from PCA, are generally utilized to analyze the common features among samples and their grouping: in fact, samples that are characterized by similar spectral signatures tend to aggregate in the score plot of the first two or three components.

k-Nearest Neighbor (k-NN) is one of the most fundamental and simple “non parametric” algorithm that is used in classification methods [71]. This algorithm has been used for creating the prediction model that is used in the present work to establish the variations of painting surfaces over time and at the different solar irradiation dose. Specifically, the proposed classification model is based on the identification of a pictorial layer before ageing, i.e., at time 0 h.

If no spectral variations occur during irradiation in Solar Box (at times: 168 h, 336 h, and 504 h), then the prediction model will identify the unchanged painting sample in all ageing times (Figure 2A). Otherwise, if Solar Box irradiation causes chemical changes in a certain sample, the prediction model will be able to identify the spectral variations in respect to time 0 and will highlight them (Figure 2B).

k-NN assumes that the data are in a feature space. More exactly, the data points are in a metric space. The data can be scalars or possibly even multidimensional vectors. Since the points are in feature space, they have a notion of distance. Each of the training data consists of a set of vectors and class label associated with each vector [72]. Given a query vector x0 and a set of N labeled instances {xi, yi} N1, the task of the classifier is to predict the class label of x0 on the predefined P classes [73]. The k-NN classification algorithm tries to find the k nearest neighbors of x0 and it uses a majority vote to determine the class label of x0. Without prior knowledge, the k-NN classifier usually applies Euclidean distances as the distance metric [74]. The performance of a k-NN classifier is primarily determined by the choice of k as well as the distance metric applied [75]. This number decides how many neighbors (where neighbors are defined by a distance metric) influence the classification. However, it has been shown that, when the points are not uniformly distributed, predetermining the value of k becomes difficult. Generally, larger values of k are more immune to the noise presented and make boundaries smoother between classes. The k-NN classification approach has been widely used in various types of classification tasks. This classification approach has gained popularity based on low implementation cost and high degree of classification effectiveness. However, its sample similarity computing is very large, which limits its applications in some cases that have high dimensional spaces or very large training sets [5,76]. In order to reduce the computation time and memory requirement without sacrificing classification capability, we apply the k-NN algorithm to the score matrix T computed with the PCA model.

## 3. Results

Preliminary results about the application of traditional techniques and HSI showed that, between the four colours examined (burnt umber, raw Sienna, Venetian red, and yellow ochre), the pigment powders mixed with gum Arabic are the most stable in regard to the artificial ageing [60]. This result encouraged to apply the same elaboration of HSI data on the entire set of samples in order to verify if effectively the pigment powders mixed with gum Arabic can be considered, in general, to be more stable in respect to the commercial products of similar colours, but of different composition. In fact, commercial mixtures contain additives and un-specified substances, added by supplier for improving the characteristics of the products, which are not present in the mixtures prepared in our laboratory only with powder and pure GA dissolved in water.

In Figure 3, the image of the sample set is shown, together with the false colour image representing the acquired raw hypercube and the selected regions of interest (ROI). The false colours in Figure 3B depend on the spectral information contained in each sample; as the data are not elaborated, it is not possible, through these false colours, to see the little differences between the pigments that in many cases are very similar in composition.

An example of ROI selection is also detailed in Figure 4. Different regions of interest were selected in order to define, for each studied pigment, a specific area to investigate.

Subsequently, SWIR acquisition was performed, and the results have been shown as average and pre-processed spectra. Pre-processing was adopted in order to better highlight the spectral differences between the paintings (Figure 5A,B). The results of PCA are displayed in Figure 5C,D.

The full colour RGB (red, green and blue) images of painting samples, at the different ageing times, are shown in Figure 6. They display the changes visible by eyes in painting samples, but these images are not able to show little colour differences that could be associated to spectral variations. The prediction/classification model, as described in the Section 2.3 named Spectral analysis, was obtained in respect to time 0 h and displayed at 168 h, 336 h, and 504 h in order to highlight the similarities or differences for each sample at the chosen ageing times. The calibration dataset of the PCA-KNN classification model has been set by considering the spectra at time 0. PCA-KNN model, applied in this modality, allows for seeing the variation of paint layers for each ageing time in respect to time 0, based on the proximity of unknowns to the different groups in the training set [60,77,78,79,80,81].

The prediction for each painting sample is obtained and reported in Figure 7, Figure 8, Figure 9, Figure 10 and Figure 11. Samples were grouped, in each figure, according to their stability, as observed by applying the prediction/classification model. The prediction map of each painting sample shows a logical (true/false) class assignment to each specific class based on strict multiple-class assignment rules. The yellow colour in Figure 7, Figure 8, Figure 9, Figure 10 and Figure 11 identifies a specific painting sample at time zero and the painting samples with the same spectral fingerprint. The blue colour is assigned to painting samples with different composition and/or painting samples that degrade during ageing. i.e., that change the spectral profile.

## 4. Discussion

The average spectra in Figure 5A show the contribution of gypsum in all samples with the SWIR absorption near 1450, 1490, and 1535 nm, and the OH/H_2_O features around 1750 and 1950 nm [44,77]. The presence, in the region 2000–2350 nm, of some absorption can be attributed to other inorganic fractions. In this region, in fact, there are the absorption of calcium carbonate (around 2230, 2341, 2373 nm) and silicates (around 2200 and 2250 nm) [78]. In some cases, CaCO_3_ is contained in the sample, as it was revealed by Fourier transform infrared (FT-IR ) spectroscopy [60].

Results of PCA are displayed in Figure 5C,D. They show a variance of 91.32% with the first three principal components. The samples are grouped in different areas of the PCA score plot according to the pigment colours and typologies. In some cases, pigments that are similar in colour and declared composition, locate themselves in different regions of the score plot, such as, for example, samples Br1 and Br2 (both being burnt umber + GA). This behaviour can also be observed for samples in pan and tube but with the same compositions, as also previously highlighted [60]. This is the case, for example, of samples GY1 and TY1 (yellow ochre in pan and tube), samples GBr2 and TBr2 (natural umber in pan and tube), GR2 and TR2 (Venetian red in pan and tube) and better in GC1 and TC1 (cobalt blue in pan and tube) and in GG2/TG2 (Viridian in pan and tube). The difference that was observed between pan and tube samples, of the same colour and typology has been attributed to the differences in additives influencing the behaviour concerning ageing [60].

Differences can be observed also between natural iron oxide pigments, similar in colour and composition, such as, for example, between the Y series (Y1-Y11). Samples Y4 and Y12 are well grouped and differentiated in the PCA score plots (Figure 5C,D, respectively) as well as samples Y5 and Y7 (Figure 5C). These results demonstrated the great potentiality of PCA in separating and grouping materials having very similar characteristics. GA and GG behaviour was widely discussed in the previous paper and are not considered here [60].

Concerning the prediction model applied on the samples set, the results give double information. Firstly, the prediction identifies the painting layer, starting from a calibration dataset composed of 58 pigments, with a low error related to pigments having similar fingerprint (i.e., similar composition). In the second step, the model evaluates the variation of the fingerprint in each sample in respect to time 0, highlighting how the spectral changes of the pigments occur during time. This double information is particularly relevant in the field of restoration, because it allows for differentiating the retouched areas in respect to the original painting, and also for monitoring the restored surfaces in order to evaluate the possible degradation during the time. Moreover, another advantage of the prediction model is the possibility for the restorer to select the retouching material with better performance in regard to ageing.

To deepen the behaviour of each single pigment, the test was applied on the different painting areas at the chosen ageing time, as shown in Figure 7, Figure 8, Figure 9, Figure 10 and Figure 11.

By observing the sample behaviour in Figure 7 and Figure 8, it can be derived that the highest stability against the artificial ageing can be seen in the following samples: GR1 (ivory black in pan), GBr1 (burnt umber 1 in pan), GR1 (Indian red 1 in pan), GR2 (Venetian red 2 in pan), GBr3 (Burnt Sienna 3 in pan), GY1 (yellow ochre 1 in pan), GG1 (bladder green 1 in pan), TR1 (Indian red 1 in tube), TBr1 (burnt umber 1 in tube), GC1 (cobalt blue 1 in pan), GG2 (Viridian 2 in pan), TG1 (chrome green 1 in tube), TG2 (Viridian 2 in tube), TC1 (cobalt blue 1 in tube), Br1 (burnt umber 1), Br4 (burnt umber 4), Br5 (burnt umber 5), Y4 (yellow ochre 4), Y5 (yellow ochre 5), Y7 (yellow ochre 7), Y8 (yellow ochre 8), Y10 (yellow ochre), R1 (red ochre 1), R2 (red ochre 2), R3 (natural umber 3), R5 (red ochre 5), Br6 (natural umber 6), Br10 (natural umber 10), and UB1 (ultramarine blue 1). These samples, in fact, exhibit no variations during irradiation times, resulting in being stable also at high total energy dose, i.e., 1.0 × 10^9^ J/m^2^ reached at 504 h of ageing. 

Samples having a similar composition seem to reduce the variability with the increasing of ageing times. For example, the spectral signature of sample GBr2 (Figure 10) tends to overlap with that of GBr1 (both burnt umber based pigments) after 336 h of ageing, corresponding to a total dose of energy equal to 6.7 × 10^8^ J/m^2^. The spectral signature of sample TR2 (Figure 9) overlaps with that of sample GR2 (both Venetian red) after 168 h of ageing (total energy applied 3.3 × 10^8^ J/m^2^). Sample TR3 (Figure 9) spectral signature overlaps with that of GR3 (both cadmium red based pigments) after 168 h of ageing. Similar behaviour can be observed in samples TBr2 and TBr1 (two natural umber watercolours in tube): the spectral signature of TBr2 (Figure 11) overlap to that of sample TBr1 after 336 h of ageing. Spectra of samples GBr4 (natural Sienna 4 in pan, Figure 9) and TBr3 (burnt Sienna 3 in tube, Figure 9) partially overlap with the spectral signature of sample Br1 (burnt umber 1) at 504 h of ageing. Another observation can be derived by observing the spectral signature of sample Y12 (yellow ochre 12, Figure 9), with the results being similar to that of Y7 (yellow ochre 7) at 336 h of irradiation in Solar box. The spectral signature of sample Y11 (yellow ochre 11, Figure 10) appears similar to that of R2 (red ochre 2) after 336 h of ageing. At last, the spectra of sample Br7 (natural umber 7, Figure 11) and of sample Br3 (natural umber 3, Figure 9) appear to be similar to that of sample Br10 (natural umber 10, Figure 8) after 336 h of ageing.

Samples GBr3 (burnt umber 3 in pan), TBr3 (burnt Sienna 3 in tube), Y9 (yellow ochre 9), and Y12 (yellow ochre 12) exhibit a definite variation after 168 h of ageing, whereas samples TR3 (cadmium red 3 in tube), Br3 (natural umber 3), and TR2 (Venetian red 2 in tube) show a partial degradation at 168 h that becomes definite at 336 h of irradiation (Figure 9). Such degradation has been associated to Arabic gum deterioration occurring between 168 h and 336 h of irradiation, combined with that of pigment components, as previously discussed [60].

Some samples undergo degradation at 336 h of irradiation in Solar box (Figure 10). Specifically, sample TBr4 (natural Sienna 4 in tube), TY1 (yellow ochre 1 in tube), GR2 (Venetian red 2 in pan), TU1 (ultramarine blue 1 in tube), and Y11 (yellow ochre 11) show stability until 168 h of ageing and then have a definite change at 336 h. Also, in this case, the changes can be associated to both gum Arabic degradation (between 168 h and 336 h, energy range 3.3 × 10^8^–6.7 × 10^8^ J/m^2^) and to the components of the watercolour and pigment mixtures.

A series of samples exhibits gradual variation until 504 h of ageing (Figure 11). In particular, samples TB1 (ivory black 1 in tube), TBr2 (natural umber 2 in tube), Y1 (yellow ochre 1), Y2 (yellow ochre 2), Y3 (yellow ochre 3), Y6 (yellow ochre 6), GU1 (ultramarine blue 1 in pan), R4 (red ochre 4), R6 (red ochre 6), R7 (red ochre 7), Br2 (burnt umber 2), Br7 (natural umber 7), Br8 (natural umber 8), and Br9 (burnt umber 9) exhibit partial degradation varying during the ageing times. It can be hypothesized that the slow variation of the spectral signature is due to the unique degradation of gum Arabic, whereas pigments seem stable and do not give clear changes at the measured time intervals.

For a better comprehension of the prediction test results, being a lot of samples, a final table is reported (Table 3).

A general assessment derived from the Table 3 is that pigment powders mixed with gum Arabic are stable to ageing apart from three samples: Br3, Y9, and Y12. This can be due to the presence of additives in the powders or to the predominance of degradation of gum Arabic that could become relevant in relation to the ratio pigment/binder.

The group of green watercolours exhibits high stability. In the case of the other watercolours, the stability varies as function of colour and of watercolour typology, i.e., pan and tube, as also previously found [60]. Sienna-based watercolours have in general low stability, both in pan and tube.

## 5. Conclusions

Hyperspectral Imaging (HSI) in the short-wave infrared was utilized to evaluate the stability to light and UV ageing of a conspicuous number of painting materials, in particular powder pigments and commercial watercolours to be used in retouching.

The new methodological approach that was chosen for monitoring the ageing behaviour of watercolour samples and pigment powder applied by gum Arabic, produced interesting results that should be further discussed and investigated. Different degradation patterns have been observed for the different pigments and also between tube and pan of the same watercolour. As previously demonstrated, gum Arabic alone clearly showed degradation occurring between 168 and 336 h of irradiation. However, when we observe the behaviour of gum Arabic/pigments mixtures, we found that it depends both on the pigment itself but also probably on the combination pigment-gum Arabic and furthermore on the presence of organic additives.

In some cases, the variation of paintings with ageing times is due to gum Arabic, such as in Y1-3, R4, R6-7, Br7-9, Br2, TB1, TBr2, and GU1 and appear as a gradually occurring phenomenon from time 0 h to 504 h of irradiation in Solar box. In this case, the combination pigment-GA seems to create a mixture that degrades slowly during time.

For other samples, the degradation of gum Arabic is the main cause of the observed variation in the prediction model, specifically in samples TBr4, TY1, TU1, GBr2, and Y11. These samples, in fact, show variations between 168 h and 336 h in the prediction model, i.e., the same range of degradation of gum Arabic.

A group of painting samples exhibits a definite change at 168 h (GBr2, TBr3, Y9, and Y12) that can be associated in part to the degradation of gum Arabic but also to that of pigment components. This time corresponds to a total solar dose of 3.3 × 10^8^ J/m^2^. In this same group we included other three samples, TR3, TR2, and Br3, whose changes are observed at 168 h but become complete at 336 h of ageing.

A conspicuous group (totally 31) of painting samples demonstrated high stability to ageing, as shown in Figure 6 and Figure 7 and Table 3, demonstrating. Fourteen of these samples are commercial watercolours and seventeen are pigment powders mixed with gum Arabic. In this case, the combination of gum Arabic, pigments, and additives (for commercial watercolours) creates stable mixtures that also prevent the degradation of gum Arabic. In fact, some authors, through surface investigations, suggested that a thin gum binder layer is present on the surface of watercolour paintings and that other components, such as pigments and additives, are located within the gum layer [82]. So, they concluded that the main changes should be attributed to gum Arabic binder. However, this result depends on pigment typology, on extender, such as calcium carbonate, and additives, which could influence the response of gum Arabic to ageing.

In general, it can be affirmed that the thirty-two investigated powder pigments mixed with gum Arabic have high or medium-high stability to ageing under simulated solar radiation, apart from three samples exhibiting low and medium-low stability and one having medium stability.

Tube and pan samples have different behaviour in relation to pigment. For example, green watercolours were demonstrated to be very stable to ageing, whereas Sienna-based mixtures have, in general, low stability. Some differences have been observed also between pan and tube of the same watercolour, such as the case of GR2 and TR2, GR3, and TR3. As previously discussed, these differences can be due to the different composition of pan and tube mixtures [60], in particular to the presence of additives in the tube watercolours necessary for obtaining the desired rheological characteristics.

The results point out the potentiality of powder pigments to be used for obtaining stable watercolours, without additives: these ones, in fact, as highlighted in other papers [19,20], are responsible for the variability and degradation in watercolours and they should be better known in order to evaluate the overall stability to ageing of these commercial materials [37,83], especially if they should be used in retouching of artworks, as commonly occurs, especially in the case of wall paintings.

As final conclusion, it can be affirmed that HSI coupled with chemometric approach allow for monitoring paint layers modifications during ageing time. Furthermore, the classification techniques based on PCA-KNN, utilizing the hyperspectral data collected by HSI, clearly outlined the potentiality of this approach for monitoring the changes occurring in the painting layers; this was possible thanks to the evaluation of little variations in the spectra during ageing times before changes can be seen by eyes. We think that this result has great relevance in the cultural heritage field because it demonstrated the possibility of detecting damages before they become irreversible. This approach could be particularly useful in monitoring artworks and restoration interventions over times at relatively low cost in respect to other analytical methods.

As future research lines, we think to apply the developed approach to other restoration materials and reintegration products based on synthetic resins that have been introduced in the conservation applications, as also suggested by conservators. The same approach in classification and predication of material behaviour in regard to ageing can be applied on protective products for cultural heritage artifacts, with the aim of testing both traditional and innovative products.

## Figures and Tables

**Figure 1 jimaging-05-00008-f001:**
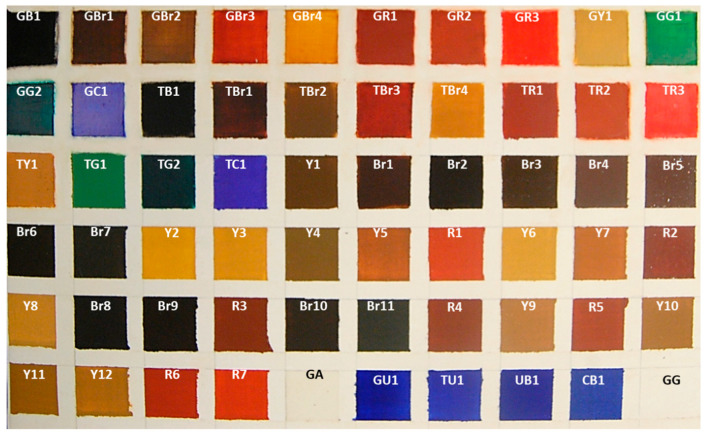
Painting samples with the corresponding abbreviations as explained in Table 1 and Table 2. GA and GG indicate gum Arabic and GG ground layer, respectively [60].

**Figure 2 jimaging-05-00008-f002:**
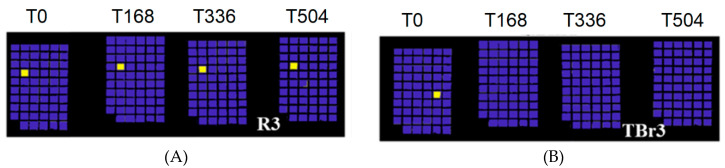
Example of paint layer (R3) which does not exhibit variations during the whole ageing cycle (**A**); example of paint layer (TBr3) showing significant spectral change after 168 h (**B**).

**Figure 3 jimaging-05-00008-f003:**
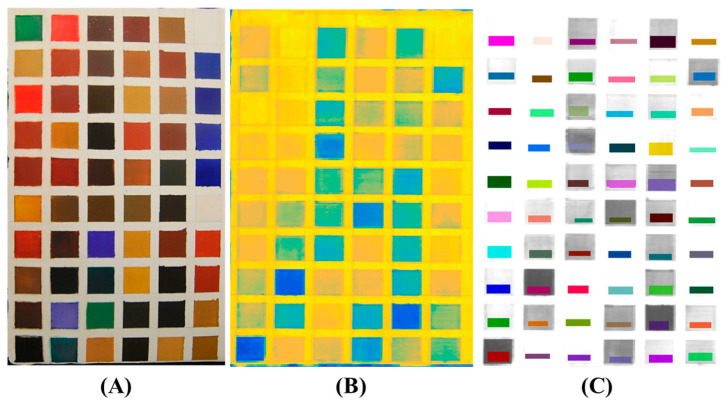
Visible image of the sample set (**A**); false colour image representing the average values of the raw hypercube after having imported it into Matlab (**B**) and the selected regions of interest (ROI) (**C**).

**Figure 4 jimaging-05-00008-f004:**
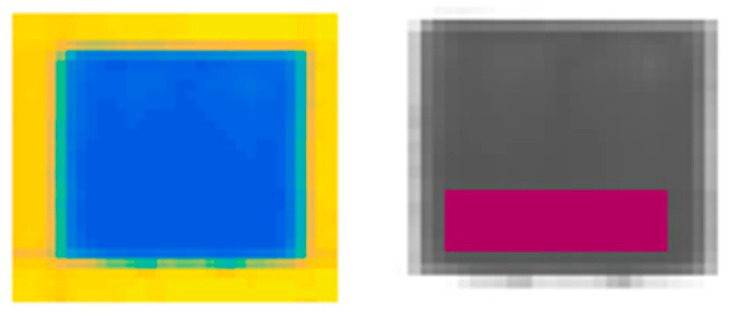
An example of ROI selection.

**Figure 5 jimaging-05-00008-f005:**
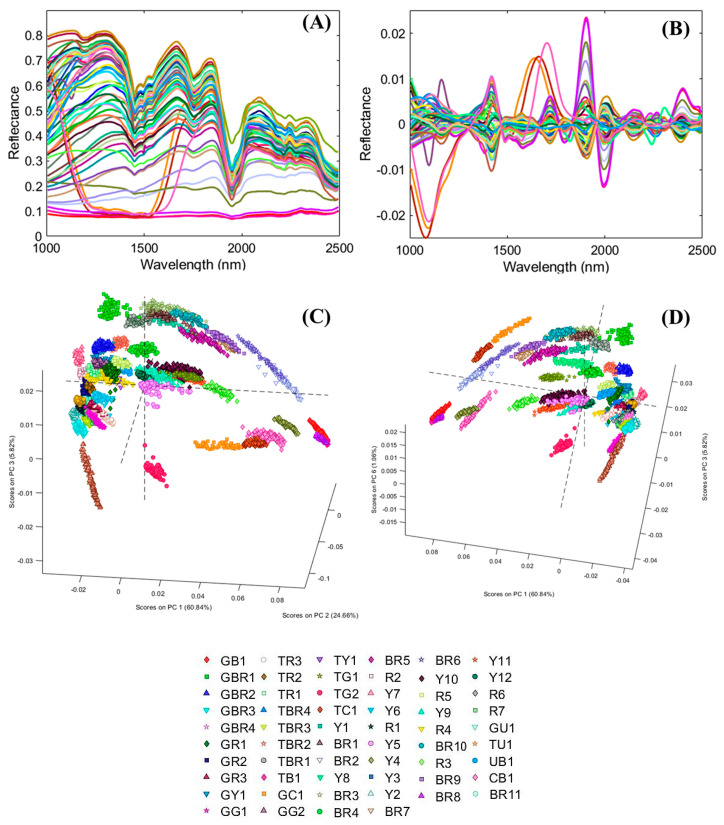
Average spectra of all painting samples (**A**); pre-processed spectra (**B**) and Principal Component Analysis (PCA) score plots (**C**,**D**).

**Figure 6 jimaging-05-00008-f006:**
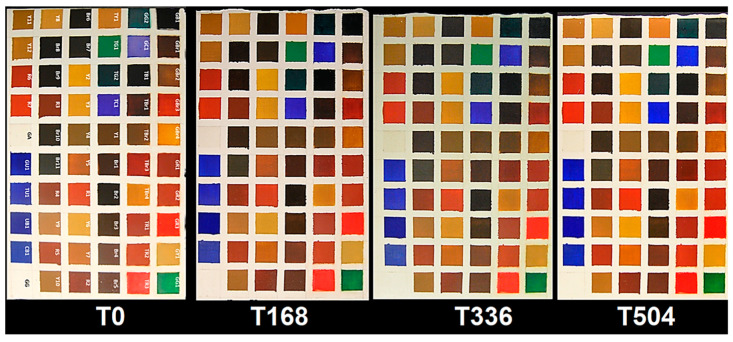
Full colour RGB (red, green and blue) images of the sample set at the different ageing times.

**Figure 7 jimaging-05-00008-f007:**
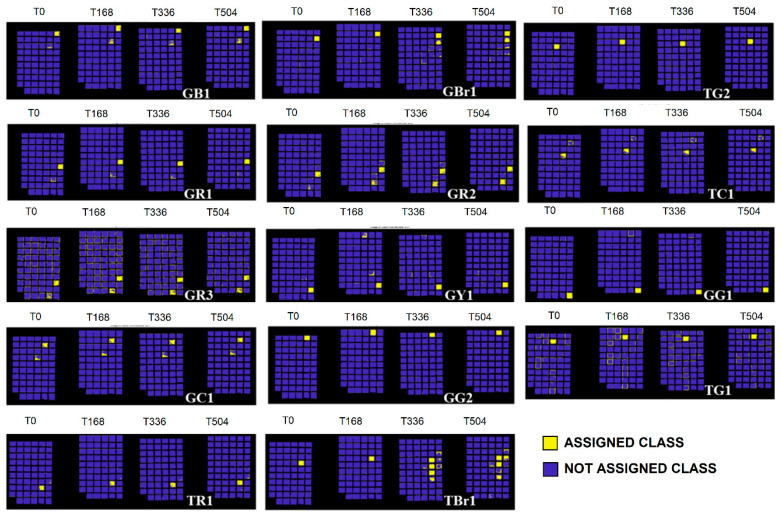
The prediction model results for commercial watercolours exhibiting no variations with ageing.

**Figure 8 jimaging-05-00008-f008:**
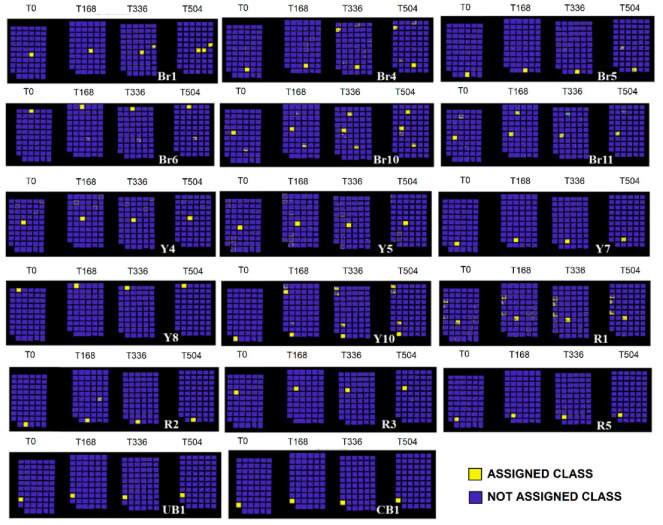
Prediction model results for pigment powders, mixed with GA, exhibiting no variations with ageing.

**Figure 9 jimaging-05-00008-f009:**
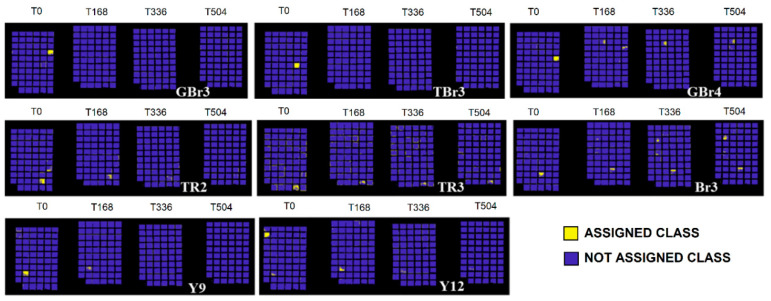
Prediction model results for painting samples exhibiting variations after 168 h of ageing.

**Figure 10 jimaging-05-00008-f010:**
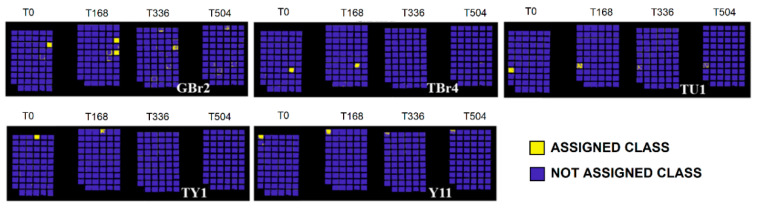
Prediction model results for painting samples exhibiting variations after 336 h of ageing.

**Figure 11 jimaging-05-00008-f011:**
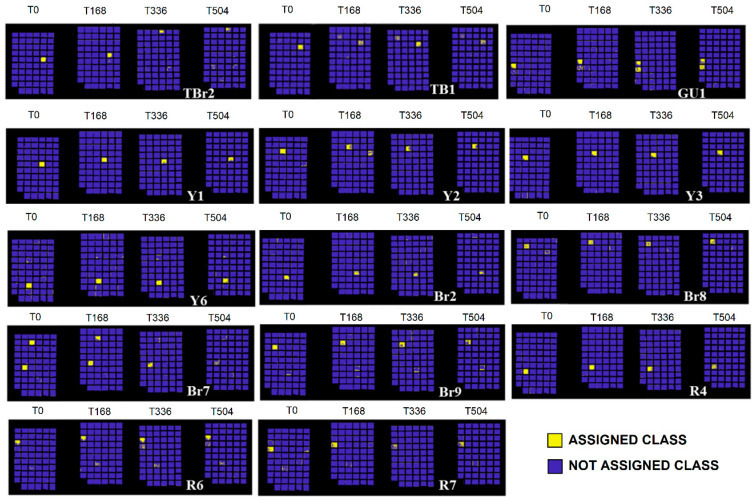
Prediction model results for painting samples exhibiting gradual variations until 504 h of ageing.

**Table 1 jimaging-05-00008-t001:** W&N samples in pan and tube, abbreviation and description.

Abbreviation	Visible Colour	Description
GB1	Black	Ivory black in pan
TB1	Black	Ivory black in tube
GBr1	Dark brown	Burnt umber in pan
TBr1	Dark brown	Burnt umber in tube
GBr2	Light brown	Natural umber in pan
TBr2	Light brown	Natural umber in tube
GBr3	Reddish brown	Burnt Sienna in pan
GBr4	Yellow-orange	Natural Sienna in pan
TBr3	Reddish brown	Burnt Sienna in tube
TBr4	Yellow-orange	Natural Sienna in tube
GR1	Dark red	Indian red in pan
TR1	Dark red	Indian red in tube
GR2	Light red	Venetian red in pan
TR2	Light red	Venetian red in tube
GR3	Light red	Cadmium red in pan
TR3	Light red	Cadmium red in tube
GY1	Light Yellow	Yellow ochre in pan
TY1	Light yellow	Yellow ochre in tube
GG1	Green	Bladder green in pan
TG1	Green	Chrome green in tube
GG2	Green	Viridian in pan
TG2	Green	Viridian in tube
GC1	Blue	Cobalt blue in pan
TC1	Blue	Cobalt blue in tube
GU1	Blue	Ultramarine blue in pan
TU1	Blue	Ultramarine blue in pan

**Table 2 jimaging-05-00008-t002:** Powder samples mixed with GA, abbreviation and description.

Abbreviation	Visible Colour	Description
Br1	Dark brown	Burnt umber in powder + GA
Br2	Dark brown	Burnt umber in powder + GA
Br3	Dark brown	Natural umber in powder + GA
Br4	Dark brown	Burnt umber in powder + GA
Br5	Dark brown	Burnt umber in powder + GA
Br6	Dark brown	Natural umber in powder + GA
Br7	Dark brown	Natural umber in powder + GA
Br8	Dark brown	Natural umber in powder + GA
Br9	Dark brown	Burnt umber in powder + GA
Br10	Dark brown	Natural umber in powder + GA
Br11	Dark brown	Natural umber in powder + GA
R1	Light red	Red ochre in powder + GA
R2	Dark red	Red ochre in powder + GA
R3	Dark red	Red ochre in powder + GA
R4	Dark red	Red ochre powder + GA
R5	Dark red	Red ochre powder + GA
R6	Light red	Red ochre in powder + GA
R7	Light red	Red ochre in powder + GA
Y1	Dark yellow	Yellow ochre in powder + GA
Y2	Light yellow	Yellow ochre in powder + GA
Y3	Light yellow	Yellow ochre in powder + GA
Y4	Dark yellow	Yellow ochre in powder + GA
Y5	Dark yellow	Yellow ochre in powder + GA
Y6	Light yellow	Yellow ochre in powder + GA
Y7	Dark yellow	Yellow ochre in powder + GA
Y8	Light yellow	Yellow ochre in powder + GA
Y9	Dark yellow	Yellow ochre in powder + GA
Y10	Dark yellow	Yellow ochre in powder + GA
Y11	Dark yellow	Yellow ochre in powder + GA
Y12	Dark yellow	Yellow ochre in powder + GA
CB1	Blue	Cobalt blue in powder + GA
UB1	Blue	Ultramarine blue in powder + GA

**Table 3 jimaging-05-00008-t003:** Sample stability evaluation derived from the prediction model.

Commercial Watercolours		Pigment Powders+Gum Arabic	
Abbreviation	Stability	Abbreviation	Stability
GB1	High	Br1	High
TB1	Medium-high	Br2	Medium-high
GBr1	High	Br3	Medium-low
TBr1	High	Br4	High
GBr2	Medium	Br5	High
TBr2	Medium-high	Br6	High
GBr3	Low	Br7	Medium-high
GBr4	Low	Br8	Medium-high
TBr3	Low	Br9	Medium-high
TBr4	Medium	Br10	High
GR1	High	Br11	High
TR1	High	R1	High
GR2	High	R2	High
TR2	Medium-low	R3	High
GR3	High	R4	Medium-high
TR3	Medium-low	R5	High
GY1	High	R6	Medium-high
TY1	Medium	R7	Medium-high
GG1	High	Y1	Medium-high
TG1	High	Y2	Medium-high
GG2	High	Y3	Medium-high
TG2	High	Y4	High
GC1	High	Y5	High
TC1	High	Y6	Medium-high
GU1	Medium-high	Y7	High
TU1	Medium	Y8	High
		Y9	Low
		Y10	High
		Y11	Medium
		Y12	Low
		CB1	High
		UB1	High

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
