# Peer review of "Hyperspectral Imaging as Powerful Technique for Investigating the Stability of Painting Samples"

_2313-433X, 2019, doi:10.3390/jimaging5010008_

Round 1

Reviewer 1 Report

The authors present a new methodological approach to evaluate the stability to light and UV ageing of a conspicuous number of painting materials, in particular powder pigments and commercial watercolours to be used in retouching using Hyperspectral imaging. The results show the potentiality of powder pigments to be used for obtaining stable watercolours, without additives. These ones, are responsible of variability and degradation in watercolours and they should be better known in order to evaluate the overall stability to ageing of these commercial materials.

The paper is well-written in terms of organization. However there are some points that should be improved.

Minor points:

- In section I), the authors should include a brief remainder of the following sections.

- In section V), the authors could include possible future research lines.

Author Response

We thank the reviewer for the positive comments on our paper.

Concerning the minor points, we corrected the manuscript accordingly.

Modifications of the manuscript have been made visible in red bold characters.

Minor points:

- In section I), the authors should include a brief remainder of the following sections.

We added a brief reminder of the sections, as required

- In section V), the authors could include possible future research lines.

In the conclusions we added some future research lines that we will develop.

Reviewer 2 Report

This paper is an interesting application of HSI. As a user of HSI in a completely different field I could not judge the importance of the work, but just reviewed the methodology, which I believe is in compliance with good scientific practice. Hence I believe this paper deserves publication.

However there are some serious readability issues that I feel are important to be addressed before publication.

lines 140-180: you carefully explain the different steps in the spectral processing, but you omit to state what the purpose of all this is. Why classify the colors? 

One general issue on the methodology is the characterization of the aging of the paint. This is not explicitly addressed in the manuscript. I get the impression it is done rather subjectively, as expressed in line 261: "by observing sample behaviour". if it is related to the prediction model, please explain a little better how this is done.

I also miss the explicit definition of the prediction model. Was that what you described in the chapter "spectral analysis|? 

Figure 5. What is expressed in the lower 4 images what do the colors mean here / color bar?

Figure 6 and 7. I have no idea how to read this. Are these similar to the lower 4 in figure 5? do they have the same range of colors? How do I interpret these images?

Now degradation is expressed as the time to degradation (for instance in line 322) The degradation is not just time dependent, but also irradiation dependent. Should it not be more fair to characterize the degradation by total solar dose (550W/m2 x exposure time in seconds, yielding a number in Joules/square meter) ?

lines 191 and further including figure 2b: figure 2b does not display a hyper-cube, it is a false color image. what do these colors mean?

Author Response

Reviewer #2

This paper is an interesting application of HSI. As a user of HSI in a completely different field I could not judge the importance of the work, but just reviewed the methodology, which I believe is in compliance with good scientific practice. Hence I believe this paper deserves publication.

Authors reply. We thank the reviewer for the positive comment about our work.

However there are some serious readability issues that I feel are important to be addressed before publication.

Authors reply. We thank the reviewer for the comments and for the  suggested changes. We modified the manuscript according to reviewer requests and we highlighted the modified text in red bold characters.

lines 140-180: you carefully explain the different steps in the spectral processing, but you omit to state what the purpose of all this is. Why classify the colors? 

Authors reply. We added in the manuscript further explanation of colours classification. This a relevant issue in conservation and restoration. In fact, it important from one hand to classify and differentiate retouched areas from the original ones, and from the other hand to monitor if retouching materials change or remain stable over time. 

One general issue on the methodology is the characterization of the aging of the paint. This is not explicitly addressed in the manuscript. I get the impression it is done rather subjectively, as expressed in line 261: "by observing sample behaviour". if it is related to the prediction model, please explain a little better how this is done.

Authors reply. The study of painting samples ageing was performed through the prediction model created by k-NN classification tool. Spectral data, obtained by Hyperspectral Imaging device operating in the SWIR range, are used for studying the surface modifications associate to pigments/binders/additives/ changes. 

The potentiality of the prediction model is to make conservators able to know the time stability of retouching materials before they could see by eye the real modifications. This is a relevant issue in conservation because the preventive knowledge of material behaviour allows for deciding which product could be used and is most appropriated for the case study.

We added further information in the discussion section of the manuscript.

I also miss the explicit definition of the prediction model. Was that what you described in the chapter "spectral analysis|? 

Authors reply: we modified the text and specified better the prediction model definition.

Figure 5. What is expressed in the lower 4 images what do the colors mean here / color bar?

Authors reply: we removed the lower part of figure 4, because it is not clear, we agree with the reviewer. Changes are better highlighted in figure 7-11

Figure 6 and 7. I have no idea how to read this. Are these similar to the lower 4 in figure 5? do they have the same range of colors? How do I interpret these images?

Authors reply: we tried to better explain figure 6, 7 (now 7 and 8) and also the following ones. In these figures each painting sample is classified with the method explained in Spectral analysis based on time 0. In this way we can compare sample spectral characteristics at time 0h, 168h, 336h and 504h. If the spectral characteristics are equal or similar to those of the sample at time 0h, the model gives the same colour. This is also a prediction method because we can see what happens at the different times before it happens.  We explained better in the manuscript figure 7-11.

The prediction map of each painting sample shows a logical (true/false) class assignment to each specific class based on strict multiple-class assignment rules. The yellow colour in Figs. 7-11 identifies a specific painting sample at time zero and the painting samples with same spectral fingerprint. The blue colour is assigned to painting samples with different composition and/or painting samples that degrade during ageing. i.e. that change the spectral profile.

Now degradation is expressed as the time to degradation (for instance in line 322) The degradation is not just time dependent, but also irradiation dependent. Should it not be more fair to characterize the degradation by total solar dose (550W/m2 x exposure time in seconds, yielding a number in Joules/square meter) ?

Authors reply: we added information about energy and improved the discussion also taking into account the energy dose on the sample surface at the different ageing times.

On the other hand, we would like to stress that the time stability is more important for conservator, as evaluation issue, rather than solar dose, because in the restoration field is more relevant to know how a material is durable over time and to try to program a possible maintenance of the restored artefacts.

lines 191 and further including figure 2b: figure 2b does not display a hyper-cube, it is a false color image. what do these colors mean?

Authors reply: we thank the reviewer for the comment. We corrected the caption and we added further details for better understand the figures and the colours.

A general English check was performed and all changes were highlighted in red bold characters.

Round 2

Reviewer 1 Report

The manuscript has been completely revised and now, the paper should be published in Journal of Imaging as is because the authors addressed all the issues.

Reviewer 2 Report

Manuscript has been improved substantially, I now agree with publication. 

One note on your remark: "On the other hand, we would like to stress that the time stability is more important for conservator, as evaluation issue, rather than solar dose, because in the restoration field is more relevant to know how a material is durable over time and to try to program a possible maintenance of the restored artefacts."

I understand that you care about time and not about a dose. yet, the time the color remains stable depends on the irradiation levels. In Greenland it would last longer than in the Sahara. Thus the dose is the best way to express the stability of the color. If you want to know how long it lasts you have to know the exposure.